# Evaluation of the Diagnostic Accuracy of the Interview and Physical Examination in the Diagnosis of Endometriosis as the Cause of Chronic Pelvic Pain

**DOI:** 10.3390/ijerph18126606

**Published:** 2021-06-19

**Authors:** Jolanta Nawrocka-Rutkowska, Iwona Szydłowska, Aleksandra Rył, Sylwester Ciećwież, Magdalena Ptak, Andrzej Starczewski

**Affiliations:** 1Department of Gynecology, Endocrinology and Gynecological Oncology, Pomeranian Medical University in Szczecin, 71-256 Szczecin, Poland; iwonaszyd@wp.pl (I.S.); sylwester.ciecwiez@pum.edu.pl (S.C.); andrzejstarcz@tlen.pl (A.S.); 2Department of Medical Rehabilitation and Clinical Physiotherapy, Pomeranian Medical University in Szczecin, 71-210 Szczecin, Poland; aleksandra.ryl@pum.edu.pl (A.R.); ptak.magda@gmail.com (M.P.)

**Keywords:** chronic pelvic pain, endometriosis, pelvic adhesions, laparoscopy

## Abstract

Background: Chronic pelvic pain affects approximately 15% of reproductive age women. It is mainly caused by adhesions (20–40%). Despite CPP being the main symptom of endometriosis, the disease is confirmed by laparoscopy only in 12–18% of cases. The aim of this study was to evaluate the results of laparoscopy in women with CCP and to assess the sensitivity and specificity of elements of an interview and clinical examination. Materials and methods: The study included 148 women with CPP. Each patient underwent laparoscopy. In laparoscopy, the presence of endometriosis and/or peritoneal adhesions was confirmed. Then, the sensitivity and specificity and the positive and negative predictive value of endometriosis symptoms or abnormalities in the gynecological examination were statistically calculated. Results: After previous surgery, adhesions were found in almost half (47%) of patients. In patients without a history of surgery, adhesions were diagnosed in 6.34% of patients. Endometriosis without coexisting adhesions was more often diagnosed in women without previous surgery (34.9%), compared to 10.58% in the group with a history of surgery (*p* < 0.05). Conclusions: Intraperitoneal adhesions are most common in women after pelvic surgery and with chronic ailments. The best results for sensitivity, specificity, positive predictive value, and negative predictive value in the diagnosis of endometriosis are found in women with irregular menstruations during which the pain increases. Laparoscopy still remains the primary diagnostic and therapeutic method for these women.

## 1. Introduction

Chronic pelvic pain (CPP) affects approximately 15% (5.7–26.6%) of women of reproductive age [1]. It is most often caused by adhesions (20–40%) or/and endometriosis (12–18%). Other causes include chlamydiosis (4.2%), irritable bowel syndrome (35%) [2], interstitial cystitis, uterine fibroids, myofascial pelvic pain (MEPP), and pelvic congestion syndrome (PCS). In a considerable number of patients, the cause of the pain cannot be found [1,3,4]. Endometriosis is a chronic disease caused by the ectopic foci of the endometrium. The most common localization of endometriosis is the ovaries, sacrouterine ligaments, vesicouterine pouch, or ovarian fossa. The main symptom of endometriosis is pain (in approximately 50% of patients), which intensifies during menstruation and ovulation. Gradually, the pain may become chronic. It is caused by a local inflammatory response and the production of prostaglandins in the foci of endometriosis, infiltration of the surrounding tissues, blood extravasation, and adhesions [3,5,6]. Depending on the location, the pain may radiate to the thigh, back, and perineum, or may be incidental to micturition and defecation. It can also increase during intercourse. However, these symptoms are not characteristic and are also present in other disease entities [7]. In gynecological examinations, in cases of pelvic endometriosis, the uterus is not very mobile, often in a retroverted position, and the sacrouterine ligaments are thickened. Sometimes micronodular lesions are found in the posterior vaginal fornix [8]. To diagnose endometriosis, imaging tests are useful. Transvaginal ultrasound is an easily-accessible, non-invasive, and relatively cheap examination [9]. In the event of endometrial cyst diagnostics, its sensitivity is approximately 86% and the specificity reaches 100%, whereas unfortunately, in the event of endometriosis located outside the ovary, the sensitivity of this method is only 38% [3,5,9]. MRI allows for a more accurate diagnosis. Whatever the protocol and MRI device, the pooled sensitivity and specificity for pelvic endometriosis diagnosis are 94% and 77%, respectively. For rectosigmoid endometriosis, the pooled sensitivity and specificity of MRI are 92% and 96% [10]. This method is particularly useful for the diagnosis of retroperitoneal foci, e.g., in the rectovaginal fascia or adenomyosis. However, it is not useful when diagnosing peritoneal endometriosis and foci less than 3 mm [11,12]. Due to the presence of bone structures, computed tomography is less useful in the diagnosis of pelvic endometriosis. The lesions that are typical for this disease do not have specific features that can be identified on MRI or CT [12]. Biochemical diagnostics is of little importance. The assessment of Ca 125 concentration has the highest sensitivity. Nevertheless, it is not a very specific marker [13]. The gold standard in the diagnosis of pelvic endometriosis is laparoscopy or laparotomy [14]. During the procedure, it is possible to assess the uterus, appendages, peritoneum, the presence of adhesions, and the size and number of foci, and thus to assess the degree of endometriosis. Treatment of endometriosis includes surgery and pharmacotherapy. Pharmacotherapy includes hormone therapy and non-steroidal anti-inflammatory drugs (NSAIDs). While postsurgical pharmacotherapy in cases of persistent pain is not controversial, it is not fully clear what the diagnostic minimum for endometriosis recognition is before the implementation of pharmacotherapy, especially hormonal treatment, as the first and often the only method of treatment. Is the mere suspicion of endometriosis enough, and if so, on what basis? Societies recommend considering the diagnosis of endometriosis on the basis of data from the history and physical examination. The diagnosis should be made on the basis of further assessment or empirical treatment. [5,15]. Which of the non-specific symptoms found in endometriosis have the highest sensitivity and specificity? Should the physical examination results be taken into consideration? Laparoscopy should be considered as a diagnostic method, since endometriosis is often among the causes of CPP. It enables causal treatment. It should be born in mind that hormonal treatment is not indifferent and is often associated with several side effects. These include nausea, weight gain, muscle spasms, depression, acne, and voice color changes; all of which of varying in intensity depending on the type of hormonal treatment used [16]. In the case of NSAID long-term use, side effects include gastric and duodenal ulcer disease, and liver and kidney damage [17,18]. It is not known what percentage of patients with “diagnosed” endometriosis are inadequately treated because of the wrong diagnosis.

Aim of the study:Assessment of the results of a laparoscopic examination of ailments in women with chronic pelvic pain in whom endometriosis was suspected based on a medical interview and clinical examination.Assessment of the sensitivity, specificity, positive predictive value, and negative predictive value of individual elements of the interview and clinical examination in the diagnosis of endometriosis, where laparoscopy was the method of verifying the diagnosis.Assessment of the causes of pain in patients with CPP, including earlier surgical procedures.

## 2. Materials and Methods

The study included 148 women hospitalized at the Department of Gynaecology, Endocrinology, and Gynaecological Oncology, Pomeranian Medical University in Szczecin, with pelvic pain, and who, based on the interview and clinical examination, had at least one of the symptoms suggesting endometriosis; such as pain increasing during menstruation, pain during intercourse, bleeding and/or pain during defecation, or micturition intensifying during menstruation, infertility, retroverted uterus and its soreness while moving, thickening of the sacrouterine ligaments, nodular lesions, and/or soreness in the pouch of Douglas. Inclusion criteria for the study were: age <55 years and chronic pelvic pain (over 6 months), as well as infertility with coexisting chronic pelvic pain. Patients who qualified for the study group reported pain lasting more than 6 months, significantly reducing their quality of life and requiring the use of periodic and permanent pain medications. On the VAS scale, the pain intensity was ≥3. Exclusion criteria were the lowering of pelvic organs above POPQ II degree, diagnosed chronic cystitis, diseases of the gastrointestinal tract (Lesniowski–Crohn disease, ulcerative colitis, diverticulitis, and colon polyposis), and diseases of the sacrolumbar spine. The study also excluded patients whose imaging tests revealed focal nodular cystic or solid lesions and those who had undergone uterine or endometrial surgery, enucleation of uterine fibroids, or urogynaecological operations related to the repair of statics of the pelvic organs. The age of the patients ranged between 19 and 53 years. The mean age was 33.39 years (SD 9.06). A detailed interview was collected for all patients. The interview included, apart from the above-mentioned criteria for inclusion in the study, colic pain, periovulatory pain, dysuric ailments, and irregular menstrual cycles. Data on past surgical procedures were also collected. The gynecological examination included symptoms constituting the inclusion criteria for the study. Diagnostic laparoscopy was performed in each patient with CPP. During laparoscopy, the reproductive organ was assessed: the uterus, appendages (ovarian, fallopian tubes), the presence of endometriosis foci on the cruciate ligaments, peritoneum, and the vesical uterine crease. If adhesions or endometriosis foci were found during laparoscopy, surgical laparoscopy was performed, during which the adhesions were released and the foci of endometriosis were removed. The size was also assessed, together with the severity of the adhesions and the amount of peritoneal fluid. The results of the study were compiled statistically by calculating the sensitivity and specificity, as well as the positive and negative predictive value of each history of endometriosis symptoms and the abovementioned deviations in the gynaecological examination. The sensitivity and specificity of the two and three most frequently coexisting symptoms in the medical history and gynecological deviations were also calculated. The aim of this was to check whether endometriosis can be more accurately diagnosed by taking into account two or three parameters at the same time. To evaluate whether adhesions are the cause of CPP, they compared symptoms in patients who had had procedures such as: caesarean section, appendix removal, or surgery on the appendages in the past and those who had no history of surgery. The second were patients who had not undergone surgery in the past. This division was aimed at assessing the presence of adhesions as the cause of CPP.

Statistical analysis was performed with the use of the licensed Statistica 13 program. The data were presented using descriptive statistics, mainly means, standard deviations, and median, as well as numbers and percentages. The chi-square test was used to evaluate the qualitative data in individual groups, and if the group was of a low size, the chi-square test with Yates’ correction was used. The normality of distribution was tested using the Shapiro–Wilk test. Quantitative data was analyzed using non-parametric Mann–Whitney U tests. The statistical significance level was *p* ≤ 0.05.

Our study was a retrospective study based on data from the patient’s medical history, physical examination, and data from the results of diagnostic laparoscopy; if necessary, operative. All patients agreed to undergo laboratory and imaging diagnostics during hospitalization and undergo surgery. All of them gave their written consent to use all the results obtained during the hospitalization for research purposes.

## 3. Results

In the entire study group of 148 examined by laparoscopy, endometriosis was found in 54 (36.48%) patients, including 23 (15.54%) with adhesions associated with endometriosis. Isolated adhesions were found in 44 (29.7%) patients. In two (1.35%) patients dilated veins were found. In 48 (32.4%) patients, no possible causes of pelvic pain were found.

The 148 patients examined were divided into those who had not undergone pelvic surgery (63) and those who had undergone surgical procedures such as appendectomy, caesarean section, and surgeries on the uterine appendages (85). The results are shown in Table 1.

In the group of patients with a positive history of surgery, only adhesions were found in almost half (47%) of the patients. In the group without a history of surgery, isolated adhesions without endometriosis were diagnosed in only 6.34% of patients. These differences were highly significant.

Endometriosis without coexisting adhesions was diagnosed more often in the group without previous surgery (34.9%), compared to 10.58% in the group with the positive history of surgery; these differences were significant at *p* < 0.05.

Significantly more often, the causes of pain were not found in women who had not undergone surgery.

Endometriosis in patients with a history of surgery was diagnosed (with and without adhesions) in 31.75% of patients, while in those who had not undergone surgery it was 42.8%. However, the differences were not significant, at *p* = 0.63330

The sensitivity, specificity, positive predictive value, and negative predictive value for individual symptoms of endometriosis occurring in the studied group of 148 patients are presented in Table 2.

None of the patients reported dysuric conditions, pain during defecation, or bleeding during micturition or from the lower gastrointestinal tract. The highest sensitivity was shown for pain increasing during menstruation, at 88.9%; the positive predictive value of this parameter was 41.38% and the negative predictive value was 81.25%. However, this parameter was characterized by low specificity, reaching 27.66%.

Other reported ailments were periovulatory, colic, and intercourse pain, and changes in a gynecological examination (nodules in the posterior vaginal fornix, thickened sacrouterine ligaments). These symptoms were characterized by low sensitivity and specificity and low positive predictive values. The number of women with these symptoms in our study was small.

The sensitivity, specificity, positive predictive value, and negative predictive value of the two most frequently coexisting symptoms in diagnosed endometriosis are presented in Table 3.

The highest specificity, reaching 98%, was found for these two simultaneously occurring symptoms: pain increasing during menstruation and infertility. At the same time, these symptoms occurring together were characterized by low sensitivity. The remaining pairs of symptoms showed moderate sensitivity and specificity. These concerned 13 patients. The most reliable, due to the number of cases (87), seem to be the results of sensitivity and specificity, as well as the positive predictive value and negative predictive value, for painful and irregular menstruations occurring simultaneously.

In the study group, pain increasing during menstruation and retroversion of the uterus occurred in 12 (8.1%) patients, pain increasing during menstruation along with nodular lesions in one patient, pain increasing during menstruation and thickened sacrouterine ligaments in four (2.7%) patients, and pain increasing during menstruation and infertility in 13 (8.78%) patients.

The sensitivity, specificity, positive predictive value, and negative predictive value of three symptoms occurring together were not calculated as there were only a few individual patients in whom three symptoms occurred simultaneously [5,16].

## 4. Discussion

Endometriosis is a disease that often manifests itself with non-specific symptoms. Ectopic foci of the endometrium are found in approximately 10% of women of reproductive age. The predominant symptom in endometriosis is pain located in the lower abdomen. However, not all lower abdominal pain is caused by endometriosis. According to Ricci et al., endometriosis is diagnosed more often in patients who report severe and prolonged pelvic pain [19]. Pelvic pain is a symptom that can also be caused by disorders of the urinary, digestive, nervous, and musculoskeletal systems. It can occur in urinary tract infections, inflammatory bowel diseases, diverticulitis, irritable bowel syndrome, minor pelvic adhesions syndrome, pelvic congestion syndrome, and fibromyalgias. It can also occur in psychosomatic disorders [7]. Pelvic congestion syndrome may also be an important cause of CPP. According to the research of Jurga-Karwacka et al., this disease was found in 12% of patients with CPP. PCS was more common in premenopausal women.

Distinguishing pain in endometriosis from its other causes is also difficult because it is similar in nature to other diseases. Not all endometriosis patients report pelvic pain. According to some authors, only 70% of patients with endometriosis diagnosed in laparoscopy reported chronic pelvic pain in their medical history [20]. Other authors estimate that endometriosis causes pelvic pain in up to 70–90% of women [21]. In our study group, of 148 patients with chronic pelvic pain and at least one symptom occurring in endometriosis, only 36.48% were diagnosed with endometriosis. Tempest et al. diagnosed endometriosis in 20% patient with CPP. In their study, most of laparoscopies carried out on young women with CPP showed no significant clinical stigmata of pelvic pain [22]. The nature of the pain seems to be of greater importance. The most common type of pain in endometriosis is pain that increases during menstruation [9]. This was found in 41.37% of patients. As a symptom in the diagnosis of endometriosis, it distinguished itself by its quite high sensitivity and specificity. Its positive predictive value was 41.38% and the negative was 81.25%. However, this symptom was also present in 45.9% of patients in whom endometriosis was not diagnosed during a laparoscopy. It seems necessary to complement the medical interview with other symptoms of endometriosis. Pain during intercourse is another common symptom among patients with endometriosis, and if it is severe (>8 on the VAS scale), it often suggests a deeply infiltrating endometriosis [23]. In their study, Ricci et al. showed a significant correlation between pain during intercourse and the diagnosis of endometriosis [9]. In our study, 25% of patients reported pain during intercourse, and only 32.4% of them had endometriosis. This symptom was characterized by low sensitivity and specificity. Its positive predictive value was 22.22% and the negative was 73.4%.

In 48 (32.4%) patients, no cause of CPP was identified. Perhaps it was retroperitoneal, deep infiltrating endometriosis, which is difficult to recognize with laparoscopy. These patients were further diagnosed in search of the causes of pain. They had, among others, a colonoscopy and magnetic resonance imaging CT scheduled. In these patients, possible causes of CPP could be, among others, intestinal diverticulosis and irritable bowel syndrome. These patients should also be diagnosed for pelvic congestion syndrome [4]. Sometimes CPP can be caused by a psychogenic reason. The large number of patients with failure to identify the cause of CPP indicates that history, physical examination, and laparoscopy are not always sufficient to explain the cause of CPP. In these patients, the diagnosis should be deepened.

It should be noted that intraperitoneal adhesions may also cause pain in the lower abdomen. In our study group, in approximately 29.73% of patients, intraperitoneal adhesions without endometriosis foci were diagnosed, and the highest number was in women who had a history of surgery within the pelvis (appendix removal, caesarean section, and surgery on the appendages). Perhaps in this group of patients a preoperative suspicion of adhesion syndrome as the cause of CPP would be more accurate than the suspicion of endometriosis. The reason for the incorrect initial diagnosis could be that the presence of intraperitoneal adhesions may imitate the presence of endometriosis, due to similar symptoms. However, in the group of 54 patients with endometriosis, adhesions were found in 42.6%, and mainly in patients (33.3%) who reported a history of surgery. Therefore, it seems that in patients with chronic lower abdominal pain who report having undergone surgery in the past, intraperitoneal adhesions should be suspected in the first place. They are the probable cause of pain in this group of patients. In these women, diagnostic laparoscopy with conversion to operative laparoscopy offers a real chance of removing the cause of chronic pain. In this case, suspecting endometriosis and therefore applying pharmacotherapy without prior peritoneal examination may be only a symptomatic treatment, and in the case of isolated adhesions without coexisting endometriosis, even a mistake. The presence of adhesions without a positive history of surgery may be caused by, for example, transfusion of pelvic inflammations. The most common cause of post-inflammatory adhesions is chlamydiosis, which is often asymptomatic. The causes of isolated adhesions should be further assessed.

The guidelines recognize the limited value of symptoms and physical examination. They are to be used to consider the diagnosis of endometriosis. In order to confirm this diagnosis, further assessment is recommended. Empirical therapies can be used to confirm endometriosis [15]. Laparoscopy is considered the gold standard in the diagnosis of endometriosis. However, it is an invasive method, and hence it is not often used as a first-line diagnostic method. During laparoscopy, we can more accurately assess the degree of advancement and obtain material for histopathological confirmation of endometriosis. During this procedure endometriosis foci and adhesions can be removed, thus reducing the patient’s pain [14].

In our studies, for two symptoms, pain intensifying during menstruation and menstrual disorders, the highest sensitivity and specificity, as well as the highest negative predictive value, occurred simultaneously. In the case of pain that intensifies during menstruation and infertility, the specificity was very high with, unfortunately, low sensitivity. Mowers EL. et al. reported that endometriosis is diagnosed in approximately 30% of infertility patients having undergone surgery. If these patients also reported painful periods in their medical history, this percentage increased to 50% [20]. Out of our patients, 75% of those with infertility were diagnosed with endometriosis.

The result of the clinical trial is also important. According to the recommendations of scientific societies, endometriosis should be suspected in patients who report continuous or periodic pain in the minor pelvis, painful menstruations, and pain during intercourse, and where a gynecological examination shows limited uterine mobility, retroversion of the uterus, thickening or soreness of the sacrouterine ligaments, or adnexal tumors [8,24]. However, as our research shows, the abovementioned symptoms do not often occur together. Some authors point to the importance of ultrasound in the preoperative diagnosis of endometriosis. Moreover, they indicate that some changes, such as adhesions, as well as tissue infiltration, may not be noticed by the surgeon during laparoscopy. They observed that not all the nodules on the sacrospinal ligament found on ultrasound were visible during laparoscopy [25].

Abott J et al. believe that, since the symptoms are diverse, varying in intensity, and not characteristic only of endometriosis, diagnosis based on medical history and physical examination is often very difficult, or even impossible. Some patients with endometriosis may not complain of CPP. How difficult it is to diagnose endometriosis based only on a medical interview is emphasized by the fact that the diagnosis is made after many years, even though the first symptoms appear in adolescence or early youth. According to literature data, there is a long delay in diagnosis from the onset of symptoms [6,26,27,28]. Currently, the time from the onset of the first symptoms to making the diagnosis is from about 5 years in Ireland and Belgium to about 10 years in Germany, Austria, and the USA [29,30,31]. This is confirmed by the fact that the symptoms reported by patients are not specific and may sometimes be so poorly marked that they do not raise the suspicion of endometriosis.

Ricci et al. attempted to diagnose endometriosis on the basis of a questionnaire in which the answers would identify patients at high risk of endometriosis. This would bring forward diagnostics and allow making a diagnosis based on symptoms. Diagnosing endometriosis based on a questionnaire that can be complemented in adolescence, allows, according to the authors of the study, accelerating diagnostics and treatment, and thus increasing the reproductive potential of the patient [20].

On the basis of the conducted research, it can be concluded that a diagnosis of endometriosis relying simply on a medical interview and physical and imaging examination is only probable. Further work is required to establish the diagnostic minimum for the diagnosis of endometriosis before beginning treatment, especially hormonal treatment, and in a much larger number of patients. At present, to make a reliable and complete diagnosis of intraperitoneal endometriosis, it still seems necessary to perform laparoscopy combined with vaporization of endometriosis foci and cutting adhesions in medical centers specializing in endoscopy. Similar suggestions are contained in the NICE report from 2017 [30].

Due to the limited number of patients, our study requires further research based on a larger number of patients.

## 5. Conclusions

In women with chronic pelvic pain and the presence of one of the symptoms suggesting endometriosis, it is diagnosed only in 1/3 of cases.Intraperitoneal adhesions are most common in women post-pelvic surgery and with chronic ailments.The highest sensitivity and negative predictive values in the diagnosis of endometriosis in women with chronic pelvic pain were demonstrated for pain that intensifies during menstruation. At the same time, the specificity for this symptom was low. The best results for sensitivity, specificity, positive predictive value, and negative predictive value in the diagnosis of this disease were found in women with irregular menstruations during which the pain increases.Laparoscopy still remains the primary diagnostic and therapeutic method in women with chronic pelvic pain and with a medical interview and physical examination suggesting endometriosis.In some patients, it is not possible to find the cause of CPP using the data from the history, physical examination, and laparoscopy. They require in-depth diagnostics to explain the cause of pain.

## Figures and Tables

**Table 1 ijerph-18-06606-t001:** The results of the laparoscopic examination in women with chronic pelvic pain and symptoms suggestive of endometriosis.

	History of Surgery n = 85	No History of Surgery n = 63	*p*
Isolated adhesions	40 (47%)	4 (6.34%)	*p* < 0.0001
Adhesions and endometriosis	18 (21.17%)	5 (7.9%)	*p* = 0.02793
Isolated endometriosis	9 (10.58%)	22 (34.9%)	*p* = 0.0032
No endometriosis or adhesions	18 (21.17%	32 (50.79%)	*p* = 0.0017

**Table 2 ijerph-18-06606-t002:** Sensitivity, specificity, positive predictive value, and negative predictive value of individual symptoms in endometriosis.

Data from the MEDICAL History and Clinical Examination		Number of Occurrences	Number of Endometriosis Cases in Laparoscopy n/%	Sensitivity	Specificity	Positive Predictive Value	Negative Predictive Value
Nature of pain	intensifying during menstruation	116	48/41.37%	88.89%	27.66%	41.38%	81.25%
	periovulatory	12	7/58.33%	53.85%	65.19%	12.96%	93.62%
	during intercourse	37	12/32.43%	32.43%	62.16%	22.22%	73.40%
	colic	12	5/41.66%	41.67%	63.70%	9.26%	92.47%
irregular periods		37	8/21.62%	41.44%	78.38%	85.19%	30.85%
infertility		16	12/75.00%	75.00%	68.18%	22.22%	95.74%
nodular lesions in the posterior vaginal fornix and its soreness		1	0/0.00%	0.00%	63.27%	0.00%	98.94%
thickened sacrouterine ligaments		4	3/75.00%	60.00%	64.34%	5.56%	97.87%
retroversion of the uterus		26	13/50.00%	50.00%	66.39%	24.07%	86.17%

**Table 3 ijerph-18-06606-t003:** Sensitivity, specificity, positive predictive value, and negative predictive value of the two most commonly coexisting symptoms in the diagnosis of endometriosis.

Data from the Medical History and Clinical Examination	Number of Occurrences	Number of Endometriosis Cases in Laparoscopy n/%	Sensitivity	Specificity	Positive Predictive Value	Negative Predictive Value
Pain intensifying during menstruation and infertility	13	7/35.84%	20.37%	97.87%	84.62%	68.15%
Pain intensifying during menstruation and irregular periods	87	41/47.12%	75.93%	51.06%	47.13%	78.69%
Pain intensifying during menstruation and retroverted uterus	12	7/58.33%	58.33%	65.44%	12.96%	94.68%
Pain intensifying during menstruation and painful intercourse	11	7/63.63%	63.64%	65.69%	12.96%	95.74%

## Data Availability

Data will be released on request.

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
