# Peer review of "Evaluation of the Diagnostic Accuracy of the Interview and Physical Examination in the Diagnosis of Endometriosis as the Cause of Chronic Pelvic Pain"

_ijerph, 2021, doi:10.3390/ijerph18126606_

Round 1

Reviewer 1 Report

The paper evaluates the diagnostic value of symptoms and clinical examination for diagnosis of endometriosis. 

The following points should be addressed before publication; 

  1. Title; the title is misleading as the main aim is to establish the diagnostic accurracy
  2. Introduction - minor suggestions 
    1. line 47 : "the symptoms" instead of "it"
    2. line 56 : add reference
    3. line 66 "assess the degree of disease severity"?
    4. line 77: rephrase the sentence
    5. line 80-82: "gestagens", why not "hormonal treatment"
    6. line 82: to my understanding, the side effects of NSAIDs are related to longterm use. 
  3. Methods: 
    1. There is no description of the laparoscopy, which areas have been examined
    2. line 113/114: rephrase
    3. infertility and its diagnosis are not included in the methods
  4. Results: overall, the results poorly relate to the objectives of the study.
    1. Line 134 : 48 (32.4%) no cause of CPP was identified. This should be clarified. One should consider that laparoscopy also does not have 100% diagnostic accuracy. Maybe these patients do have endometriosis, which would significantly change the results and conclusions of the study. In any case, these patients should be further assessed and explained. The same conclusion can be made for the patients with isolated adhesions, without details on the surgical procedure, the conclusions and lap diagnosis can not be verified.
    2. The first analysis (line 135 and following) on the number of previous surgeries as diagnostic criteria was not included in the objectives. I am also doubting the value.
    3. The symptom severity is not mentioned, nor included in the analysis
  5. Conclusion 
    1. The conclusion needs to be thoroughly reassessed. The authors suggest that guidelines on endometriosis (such as those from ESHRE) state that endometriosis is diagnosed with symptoms. This is incorrect. The guidelines acknowledge the limited value of symptoms/exam only and state that diagnosis should be considered, and do recommend further assessment, or verification through empirical treatment. This is not correctly represented in the conclusion. The authors seem to be convinced that laparoscopy is the gold standard for endometriosis diagnosis, which probably caused them to overlook the issues with laparoscopy. These are not described in the text nor methods. The high number of undiagnosed cases is also not further discussed. 
    2. the limited sample size is not mentioned in the conclusion
  6. Further sentences that could benefit from rephrasing; 
    1. line 195, 165, 168

Reviewer 2 Report

Dear Authors.
I would like to thank you for the opportunity to review the article "THE REASONS OF CHRONIC PELVIC PAIN IN THE LIGHT OF LAPAROSCOPIC EXAMINATION IN PATIENTS WITH ENDOMETRIOSIS SUSPICION". The concept is interesting, the planning of the study seems to be correct. As we know, chronic pelvic pain is a problem faced by female coats in the reproductive period and significantly affects their quality of life and the quality of sexual life. Proper diagnosis of the cause significantly facilitates treatment planning, and thus therapeutic success.
The manuscript needs improvement:
1. Lack of the approval number of the Bioethics Committee and the date of its issue.
2. The description of the statistics is very modest and makes the manuscript illegible when the reader analyzes the results.
3. Currently, there are many publications on chronic pelvic pain related to pelvic congestion syndrome (eg doi: 10.1371/journal.pone.0213834) about which the authors did not comment (even in the discussion), successfully treated with minimally invasive techniques. I recommend supplementing the manuscript with this issue.

Sincerely...

Round 2

Reviewer 1 Report

Line 119 - the use (space missing)

Line 140 - the size was 

Line 148 - Thank you for clarifying the relevance of previous surgeries. However, I would not state that the patients were divided in 2 groups, but rather that 'to evaluate the presence of adhesions as the cause of CPP, patients who have undergone previous surgeries such as  ..., were compared to ...